# Spontaneous Closure of the Arterial Duct after Transcatheter Closure Attempt in Preterm Infants

**DOI:** 10.3390/children8121138

**Published:** 2021-12-05

**Authors:** Mathilde Méot, Raymond N. Haddad, Juliana Patkai, Ibrahim Abu Zahira, Anna Di Marzio, Isabelle Szezepanski, Fanny Bajolle, Elsa Kermorvant, Alexandre Lapillonne, Damien Bonnet, Sophie Malekzadeh-Milani

**Affiliations:** 1Centre de Référence Malformations Cardiaques Congénitales Complexes—M3C, Hôpital Necker-Enfants Malades, 75015 Paris, France; raymond.haddad@aphp.fr (R.N.H.); ibrahim.abuzahira@aphp.fr (I.A.Z.); isabelle.szezepanski@aphp.fr (I.S.); fanny.bajolle@aphp.fr (F.B.); damien.bonnet@aphp.fr (D.B.); guiti.milani@gmail.com (S.M.-M.); 2Neonatology Department, Port-Royal Hospital, 75014 Paris, France; juliana.patkai@aphp.fr; 3Anesthesiology, Hôpital Necker-Enfants Malades, 75015 Paris, France; anna.dimarzio@aphp.fr; 4Neonatology Department, Hôpital Necker-Enfants Malades, 75015 Paris, France; elsa.kermorvant@aphp.fr (E.K.); alexandre.lapillonne@aphp.fr (A.L.); 5UFR de Médecine, Université de Paris, Site Cordeliers, 75006 Paris, France

**Keywords:** arterial duct, closure, premature

## Abstract

(1) Background: Transcatheter closure of the patent arterial duct (TCPDA) in preterm infants is an emerging procedure. Patent arterial duct (PDA) spontaneous closure after failed TCPDA attempts is seen but reasons and outcomes are not reported; (2) Methods: We retrospectively included all premature infants <2 kg with abandoned TCPDA procedures from our institutional database between September 2017 and August 2021. Patients’ data and outcomes were reviewed; (3) Results: The procedure was aborted in 14/130 patients referred for TCPDA. Two patients had spasmed PDA upon arrival in the catheterization laboratory and had no intervention. One patient had ductal spasm after guidewire cross. Four patients had unsuitable PDA size/shape for closure. In seven patients, device closure was not possible without causing obstruction on adjacent vessels. Among the 12 patients with attempted TCPDA, five had surgery on a median of 3 days after TCPDA and seven had a spontaneous PDA closure within a median of 3 days after the procedure. Only the shape of the PDA differed between the surgical ligation group (short and conical) and spontaneous closure group (F-type); (4) Conclusions: In the case of TCPDA failure, mechanically induced spontaneous closure may occur early after the procedure. Surgical ligation should be postponed when clinically tolerated.

## 1. Introduction

The management of the patent arterial duct (PDA) in premature (<28 weeks gestational age) and extremely low birth weight (1500 g) infants remains controversial [1,2]. Indication to close a PDA in premature neonates is debated whichever method and timing are used. In the last decade, there has been a trend towards a conservative approach in the treatment of PDA in preterm infants to avoid adverse effects of pharmacological agents and of surgical ligation. Indeed, the majority of arterial ducts close spontaneously [2]. However, a subgroup of patients, mainly extremely premature patients (<26 weeks gestational age), will suffer from consequences of a hemodynamically significant PDA in the first days or weeks of life, and will likely benefit from intervention. Percutaneous closure is now considered safe, with high rates of success [3,4,5] even in infants <1000 g [6]. Still, there remain concerns about traumatic tricuspid regurgitation [7,8], iatrogenic left pulmonary artery (LPA) stenosis and coarctation of the aorta [9], and device embolization or migration, among other more anecdotic complications [10]. These rare complications can be life threatening and consensus guidelines have recently been published to prevent them, and to treat them in case of occurrence [11]. The PDA is known to be an unpredictable vessel, with closure and reopening occurring from multiple and complex stimuli. Transitory closure of the arterial duct followed by a reopening during a sepsis or multiple drug course has been observed [12,13]. Indeed, functional constriction should be followed by anatomic remodeling, resulting in the formation of the ligamentum arteriosum [1]. Here, we describe the patients for whom the transcatheter PDA closure procedure (TCPDA) was stopped, the reasons to abandon the procedure, and the outcome of this specific population. We focus our discussion on the patients in whom mechanical stimulus of the PDA led to subsequent definitive closure.

## 2. Materials and Methods

### 2.1. Study Population

All premature infants weighing less than 2 kg and referred to our institution for TCPDA since September 2017 were prospectively included in the Premiclose registry before the procedure. Parents’ informed consent for the procedure and for inclusion in the Premiclose registry was obtained at admission in the NICU (IRB/Local Authorities Agreement: 2149450V0). We retrospectively reviewed all abandoned procedures and the outcomes of these patients. Data collected included gestational age and birth weight, procedural age and weight, all details regarding medical history before the procedure with particular attention to complications of PDA (intra ventricular hemorrhage, necrotizing enterocolitis, bronchodysplasia, pulmonary hemorrhage, renal failure), and attempts to close PDA with medical treatment. We also collected all procedural data and outcomes.

### 2.2. Procedure

Most of our patients are referred by NICUs throughout the Grand Paris area. They are admitted to the NICU the day before the procedure. Echocardiogram is performed to assess size, length, and morphology of the PDA, its relationship with the great vessels, and the diameters of the aorta and of the LPA. We also ensure that the PDA is hemodynamically significant, with a clinical impact and echocardiographic features of pulmonary overflow and systemic hypoperfusion. We make sure that there are no contra-indications to the procedure such as active infection or severe pulmonary hypertension. The ultrasound scan is systematically repeated when the patient is in the catheterization laboratory before femoral vein puncture. The procedure is performed under general anesthesia. We have previously reported our technique for TCPDA [8]. In our early experience, we used to perform angiograms to assess PDA diameter and length, but for the last 2 years, the device size has been chosen only on echocardiography. We exclusively use the Amplatzer Piccolo Occluder (Abbott Structural Heart, Plymouth, MN, USA), a device designed to close typical type F ductus of premature and/or extremely low birth weight neonates with FDA approval. The device is delivered intraductally and correct position is assessed by fluoroscopy and echocardiography. Before releasing the device, we check with ultrasound the absence of residual shunt, as well as complete patency of LPA or aortic isthmus. Unacceptable results are maximal velocity above 2 m/s, and/or typical obstructive color flow Doppler in the LPA or in the aortic isthmus. At the end of the procedure, the patient is transferred to the NICU and is monitored for at least 24 h before transfer to the referring hospital. Echocardiogram is performed on the evening of the procedure and the day after.

### 2.3. Statistical Analysis

Categorical variables are given in proportions (percentage). Continuous variables are summarized by median and interquartile range (IQR), minima and maxima.

## 3. Results

Fourteen procedures were abandoned and included in this series. The global population was 130 patients. Median age and weight at birth in children for whom the procedure was stopped were 25.3 GA and 850 (IQR 625–937.5) grams, respectively. Median age and weight at procedure were 32.5 (23.25–37) days and 980 (900–1378) grams, respectively. All patients had received pharmacological treatment prior to procedure (ibuprofen or paracetamol, at least two courses) without success. Two/14 patients had severe intraventricular hemorrhage (grade ≥ 3 with ventricular dilation), one had necrotic enterocolitis requiring surgery, and none had pulmonary hemorrhage.

Transitory closure of the arterial duct was not observed before TCPDA attempt. Reasons of stoppage of the procedure are detailed in Table 1 and Table 2. Ductal spasm occurred before the beginning of the procedure in two patients, and in one patient ductal spasm was observed and persisted after guidewire crossing of the PDA. In four patients, the size or shape of the arterial duct was not suitable for percutaneous closure (too short and/or too large or conical). In seven patients, the device was deployed but retrieved because an obstruction on the LPA or on the aorta was present despite several attempts to reposition the device properly, and a significant residual shunt was unacceptable when a smaller device was deployed.

Outcome can be divided in two groups: surgical ligation or spontaneous closure. Five patients needed surgery (Figure 1 and Table 2). Surgical ligation was performed after a median delay of 3 days after the catheterization by left posterolateral thoracotomy. One patient had a pericardial effusion after surgery and required a drainage. Amongst the two patients with a spasmed PDA upon arrival in the catheterization laboratory, one PDA reopened the day after, and finally closed after a third cure of ibuprofen. In the seven remaining patients, the arterial duct spontaneously closed after a mean delay of 3 days following TCPDA attempt (Figure 1 and Table 1).

We did not notice major differences in clinical characteristics between the two groups (surgical ligation versus spontaneous closure) except for the shape of the PDA: three out of five patients in the surgical group had a short duct, among whom two had a conical rather than tubular duct (Table 2). Only one out of seven patients in the spontaneous closure group had an unusually short but still tubular duct. The remaining six patients had a classical F-type duct. As shown in Table 3, in which outcomes are presented according to the year of the procedure, and in light of our experience of spontaneous closure of the PDA after failure of percutaneous closure, no patient had been operated on in the last 18 months.

One patient particularly caught our attention (Figure 2): a 780 g infant with a 3 mm PDA (length 6 mm) had a 4*2 mm Piccolo device placed in the PDA. An aortic coarctation was observed on TTE and the device was removed. The arterial duct spasmed during the procedure as well as the aortic isthmus and a transient typical coarctation Doppler flow persisted after removal of the device. The aortic isthmus and the arterial duct both reopened, and Doppler flow in the aortic isthmus normalized within a few minutes. The child was transferred to the NICU, and we observed spontaneous closure of the PDA after 3 days without associated aortic coarctation. 

None of the patients of this cohort died.

## 4. Discussion

TCPDA in preterm infants is an emerging procedure that will become or is becoming a preferred choice in centers with experienced interventionists and well-trained cardiac sonographers. Limited risks and relatively low rates of complications are described with this procedure in experienced hands [11], but complications can be disastrous and should be avoided at any cost including stoppage of the procedure. In our cohort, we describe three reasons for procedural interruption. Spasm of the arterial duct before the start of the procedure or after crossing the PDA with a guidewire confirmed the well-described spastic nature of the arterial duct. In five patients, spastic duct was noticed at the beginning or during the procedure. It is not infrequent in practice to see transitory ductal spasm, or changes in the ductal size between admission in NICU and the transfer in the catheterization laboratory. Oxygen administration during induction or anesthetics, as well as the child’s possible hemodynamic or respiratory lability, may contribute to ductal spasm. Whether these spastic ducts are more prone to spontaneous closure with or without stimuli remains an open question. Other studies have reported the spasm of PDA in infants and children during TCPDA [14]. This event was considered undesirable during the procedure as it compromised its success by precluding the choice of an adequate device, and increased the risk of subsequent embolization [15]. Formerly premature children, even later in infancy, seem to be more prone to ductal spasm [14,15,16]. The second reason to abandon the procedure was related to the anatomy of the PDA that was not suitable for percutaneous closure with the Piccolo Occluder. These cases were observed at the beginning of our experience with TCPDA in premature babies and are no longer observed as careful selection of patients has now improved. The last reason to stop the procedure was related to LPA or aortic obstruction after multiple attempts to reposition or change the size of the device. In our department, we considered that even mild obstruction of the LPA was unacceptable and that the device should not be released. These cases tend to be rarer with experience but are still observed in our current practice.

All patients survived and followed two different paths. Five had surgical closure and the arterial duct closed spontaneously in the remaining seven patients. In patients with delayed spontaneous closure of the PDA, the main reason for procedural stoppage was LPA or aortic coarctation or spasm. In the surgical group, the main reason for stopping the procedure was unsuitable PDA anatomy with large, conical, or short ducts. It is of note that spontaneous closure occurred after a mean delay of 3 days after the PDA percutaneous closure attempt, and that, in the surgical group, three patients were operated on within 48 h after the cardiac catheterization. Today, we would advise to defer the surgical closure if the child’s condition allows waiting for three more days as spontaneous closure may occur until the third day following the cardiac catheterization.

The arterial duct normally closes a few hours or days after birth in term neonates, and this closure completes the transition from the fetal circulation to the extra-uterine circulation. The closure of the arterial duct occurs in two phases: first there is a functional closure of the lumen with smooth muscle constriction. This functional closure produces a hypoxic–ischemic zone in the vessel wall that, afterwards, triggers anatomical occlusion and remodeling by cell death and vascular fibrosis [17]. Several mechanisms explain the non-closure of the arterial duct in premature neonates, such as a higher sensitivity to prostaglandin and nitric oxide, a reduction in prostaglandin catabolism in the lung in early gestation, and the lower density of oxygen sensing ion channels. The histology of the arterial duct in preterm babies is different from that of term babies. Indeed, the intimal cushions are hardly formed, and the media of the arterial duct contain fewer layers of contractile smooth muscle cells with incomplete vasa vasorum [18]. This can result in partial postnatal constriction that is insufficient to generate the hypoxic zone and the ischemia-driven remodeling that is critical for permanent closure. It also explains the propensity of the arterial duct to reopen in premature babies [12,13].

While pharmacologically induced closure of the arterial duct has been widely studied, mechanically induced closure has not been studied. The consequences of the traumatism of the inner surface of the arterial duct with a guidewire or a device and/or the acute stretch provoked by the deployment of the occluder in the lumen might activate the process of spontaneous closure. This is purely speculative as the occurrence of this event remains unknown. An important issue is the delay of closure after the procedure. We observed in this short series of “failed” TCPDA that the delay of spontaneous closure was three days. If the hemodynamic status of the child and the risks of prolonging the intensive care duration are acceptable, we would advise to wait 72 h before referring the child for surgical ligation of the PDA. Another interesting question would be to attempt pharmacological closure after “failed” TCPDA if there is no contra-indication for the available drugs. It is indeed conceivable that the synergy of mechanical and biochemical factors would potentialize the closure of the arterial duct. Considering that the more serious adverse events related to TCPDA in premature infants result from the placement of the device (left pulmonary artery stenosis, coarctation, embolization), we think it can be justified to attempt percutaneous closure even in challenging situations (e.g., extremely small babies between 700 and 800 g with very large arterial ducts and small adjacent vessels) [19]. Failing to properly position the device is a rare event, and it is probably more ethical to indicate surgical ligation in the anatomical conditions for which failure is highly predictable. Here, we wanted to warn our interventionist colleagues and neonatologists involved in TCPDA programs of this unexpected favorable outcome. We show that the mechanical stimulation of the ductal tissue may induce a delayed permanent closure within a relatively short delay. As this may happen when the arterial duct is a typical F-type duct, we believe that an imperfectly positioned device should not be released. We also think that “failed” procedures should be reported to accumulate experience and potentially explore alternative strategies for PDA closure when the available devices cannot be safely released. Large multicenter registries are needed to evaluate if this observation is anecdotical or more frequent than we might think.

## 5. Conclusions

In the case of failure of TCPDA, a mechanically induced spontaneous closure may occur a few days after the procedure. We suggest that surgical ligation should be postponed if this is medically reasonable. This delayed closure should also have an impact on the decision to release the device if it is imperfectly positioned. Sharing data on this unusual outcome might be interesting to promote complementary strategies in this field such as combining mechanical stretch and pharmacological treatment to promote or accelerate PDA closure.

## Figures and Tables

**Figure 1 children-08-01138-f001:**
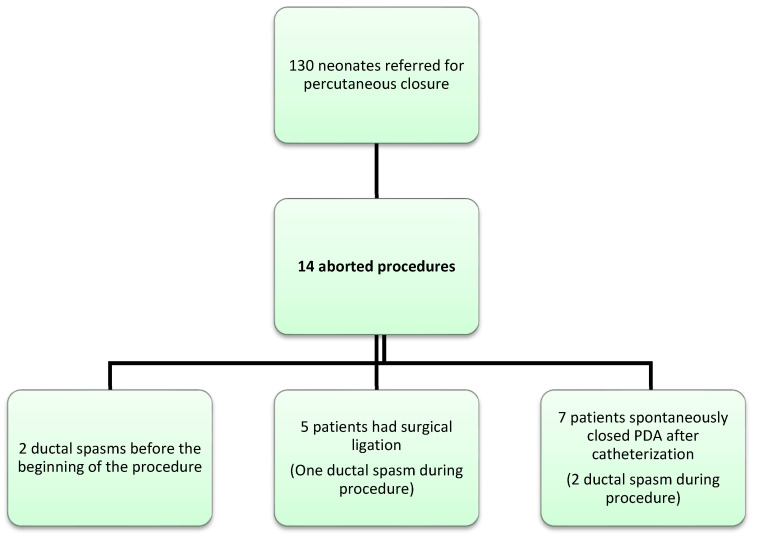
Study flow chart.

**Figure 2 children-08-01138-f002:**
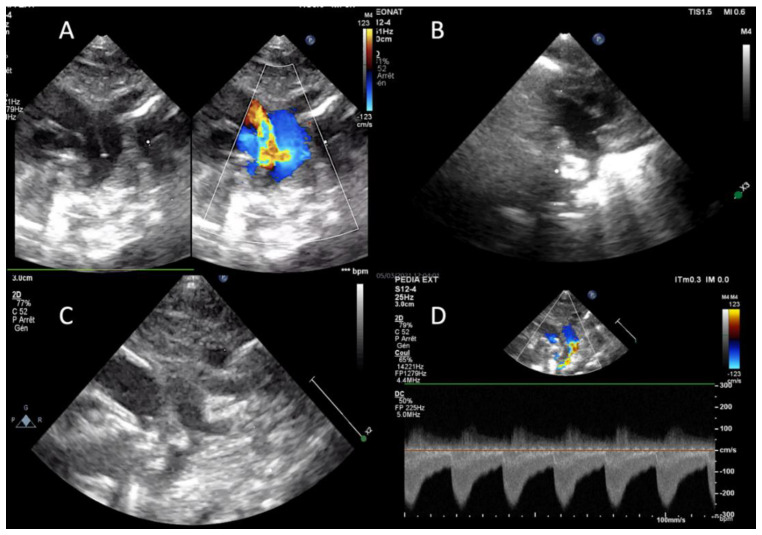
(**A**) PDA and aortic isthmus at the beginning of the procedure. Arrow is the F-type large PDA. (**B**) Protrusion of the device in the aorta occluding the isthmus. (**C**) Aortic isthmus and PDA spasm persisting after removal of the device. (**D**) Typical coarctation flow with diastolic run off on the isthmus after removal of the device.

**Table 1 children-08-01138-t001:** Characteristics of the seven patients with patent ductus arteriosus (PDA) permanent spontaneous closure after attempted percutaneous closure.

	GA ^1^ (weeks)	BW ^2^ (gr)	PA ^3^ (days)	PW ^4^ (gr)	PDA Type and Size at PA end ^5^ on TTE (mm)	Devices Tried ^6^	Reason to Abandon Procedure	Delay between Catheterization and Spontaneous Closure (days)
**1**	27 + 1	1010	37	1685	F/3	No device	Spasm Too small	0
**2**	24 + 2	515	36	875	F/2.7	4*2	Coarctation	1
**3**	26	915	29	1370	F/3	5*2 4*2	LPA stenosis	6
**4**	24 + 3	640	24	780	F/3	4*2	Coarctation Spasm	3
**5**	24 + 2	610	28	890	F (short 6 mm) 3.5	5*2	Coarctation	5
**6**	25 + 1	960	17	1100	F/3	4*2	LPA and aorta stenosis	Small, not hemodynamically significant
**7**	25 + 6	850	21	1000	F/4	5*2	Unstable device	2

^1^ GA: gestational age; ^2^ BW: birth weight; ^3^ PA: procedural age; ^4^ PW: procedural weight; ^5^ PA end: pulmonary artery end of the arterial duct; ^6^ Device tried: size of the Piccolo Occluder used for the closure attempt.

**Table 2 children-08-01138-t002:** Characteristics of the five patients who needed surgical ligation after attempted percutaneous closure.

Patient	GA ^1^ (weeks)	BW ^2^ (gr)	PA ^3^ (days)	PW ^4^ (gr)	PDA Type and Size at PA End ^5^ on TTE (mm)	Devices Tried ^6^	Reason to Abandon Procedure	Delay between Catheterization and Surgery (days)
1	26 + 2	1000	38	1400	F/3.7	5*2	Spasm LPA stenosis	1
2	23 + 3	765	18	900	F (but short)/4	5*2	Unstable device	4
3	25 + 6	855	37	1425	A/4.4		Too large on angiography	0
4	23 + 4	600	37	960	F/3.2	5*2 4*2	LPA stenosis	3
5	27	NA	37	900	A/4.5	5*2	Too large	6

^1^ GA: gestational age; ^2^ BW: birth weight; ^3^ PA: procedural age; ^4^ PW: procedural weight; ^5^ PA end: pulmonary artery end of the arterial duct; ^6^ Device tried: size of the Piccolo Occluder used for the closure attempt.

**Table 3 children-08-01138-t003:** Table representing the evolution of the stopped procedures over the years.

Year	Spasm of the PDA in the Catheterization Lab	Surgical Closure of the PDA	Delayed Spontaneous Closure of the PDA
2017	0	1	1
2018	1	2	0
2019	1	1	2
2020	0	1	2
2021	0	0	2

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
