# Peer review of "Spontaneous Closure of the Arterial Duct after Transcatheter Closure Attempt in Preterm Infants"

_children, 2021, doi:10.3390/children8121138_

Round 1

Reviewer 1 Report

The authors present a retrospective review of a single-center experience with spontaneous closure of PDA after attempted transcatheter closure.  Unfortunately, this population size is very low and hard to determine clinical significance.

I think the numbers need to be improved and this should be done as a registry study or multicenter study.  

Author Response

Dear reviewer,

We thank you for the review of our paper and for your comments.

Reviewer’s comment #1: The authors present a retrospective review of a single-center experience with spontaneous closure of PDA after attempted transcatheter closure.  Unfortunately, this population size is very low and hard to determine clinical significance. I think the numbers need to be improved and this should be done as a registry study or multicenter study.  

Dear reviewer, we agree with your comments, numbers are low and large registries are needed to draw robust conclusions about these findings. We wanted to report on this finding because we initially thought that this was anecdotal. But as we observed several cases with the same outcome, we thought it would be interesting to report them, highlighting on the need to report those cases to have more evidence on this event, whether it is anecdotal or more frequent than what we think.

We modified the text and add a sentence in that direction emphasizing on the need of large registries to draw robust conclusions (line 229).

We thank you for your comment and have modified the manuscript accordingly. We hope that the modifications will make the manuscript suitable for publication.

Reviewer 2 Report

Méot and colleagues report on a case series of spontaneous closure of PDA after failed attempt at catheter closure. Their experience is relevant given the rapidly evolving field of interventions to close the hemodynamically significant PDA.

The Abstract appropriately summarizes the salient content of the paper.

The background information is adequate, as is the description of the methods, including ethical considerations for this research.

The Results are reasonably detailed, as desirable in this type of report.

Table 2 should be formatted to fit within 1 page.

Figure 2 is acceptable, but the data might be best (and most efficiently) represented in a table. Also, there should be a zero for the delayed spontaneous closure in 2018. In any case, the data and title/legend should be on the same page.

In the Discussion, I believe it is premature to characterize TCPDA as a gold standard, since there is no high-quality evidence comparing it to alternative treatments. It might be fair, however, to call it a "preferred choice".

The authors appropriately acknowledge the speculative nature of the mechanically-induced spasm hypothesis. However, they might wish to venture an alternative speculation for the couple of spastic PDAs at the beginning of the catheterization procedure.

The references are adequate for the material covered in the manuscript.

Minor details:
Line 38, "<26GA", should be more formerly expressed as <26 weeks gestational age, or <26 weeks’ gestation.

Line 41, "remains" should be "remain"

Line 97, “cures” should be “courses”

Line 180, “differ” should be “defer”

Author Response

Dear reviewer,

We thank you for the review of our paper and thank you for all your comments.

Please find the response and modifications done according to your comments.

We thank you for your comment and have modified the manuscript accordingly. We hope that the modifications will make the manuscript suitable for publication.

Comment #1: Table 2 should be formatted to fit within 1 page.

We agree with you and we made the modifications accordingly.

Comment #2: Figure 2 is acceptable, but the data might be best (and most efficiently) represented in a table. Also, there should be a zero for the delayed spontaneous closure in 2018. In any case, the data and title/legend should be on the same page.

Thank you for your comment. This figure has been removed and and replaced by a table according to your comments, as well as the legends.

Comment #3: In the Discussion, I believe it is premature to characterize TCPDA as a gold standard, since there is no high-quality evidence comparing it to alternative treatments. It might be fair, however, to call it a "preferred choice".

Dear reviewer, we totally agree with this comment, and this has been modified in the texte (line 155).

Comment #4: The authors appropriately acknowledge the speculative nature of the mechanically induced spasm hypothesis. However, they might wish to venture an alternative speculation for the couple of spastic PDAs at the beginning of the catheterization procedure.

We agree with this comment and have added this sentence in the text:

In 5 patients, spastic duct was noticed at the beginning or during the procedure. It is not infrequent in practice to see transitory ductal spasm, or changes of the ductal size between the admission in NICU and the transfer in the cath lab. Oxygen administration during induction or anesthetics, as well as possible child’s hemodynamic lability, may contribute to ductal spam.  Wether these spastic ducts are more prone to spontaneous closure with or without stimuli remains an open question.

Comment #5: Line 38, "<26GA", should be more formerly expressed as <26 weeks gestational age, or <26 weeks’ gestation.

This has been modified in the text.

Comment #6: Line 41, "remains" should be "remain".

This has been modified in the text.

Comment #7: Line 97, “cures” should be “courses”.

This has been modified in the text.

Comment #8: Line 180, “differ” should be “defer”.

This has been modified in the text.

We thank you for your comment and have modified the manuscript accordingly. We hope that the modifications will make the manuscript suitable for publication.

Reviewer 3 Report

The authors sought to evaluate the spontaneous closure of a PDA after failed transcatheter closure in a cohort at their institution.   Comments/recommended revisions:

  • The wording on page 2 line 46 is slightly unclear and may just need a slight change: "Fetal PDA is known to be an unpredictable vessel, which closure and reopening result from multiple and complex stimuli." Perhaps it should read as "The PDA is known to be an unpredictable vessel, with closure and reopening occurring from multiple and complex stimuli."
  • I thought that the Discussion was very insightful with a clear description of the situation and recommendations to wait 3 days for spontaneous closure, if possible.

Author Response

Dear reviewer,

We thank you for the review of our paper and for your comments.

Please find the response and modifications done according to your comments:

Reviewer’s comment #1: The wording on page 2 line 46 is slightly unclear and may just need a slight change: "Fetal PDA is known to be an unpredictable vessel, which closure and reopening result from multiple and complex stimuli." Perhaps it should read as "The PDA is known to be an unpredictable vessel, with closure and reopening occurring from multiple and complex stimuli."

We agree with your comment and have modified the text accordingly.

We thank you for your comment and we hope that the modifications will make the manuscript suitable for publication.

Round 2

Reviewer 1 Report

The authors added wording to address the concern about low sample size and added focus on the need for additional/multicenter studies.